# Taste-Driven Responsiveness to Fat and Sweet Stimuli in Mouse Models of Bariatric Surgery

**DOI:** 10.3390/biomedicines10040741

**Published:** 2022-03-22

**Authors:** Aurélie Dastugue, Cédric Le May, Séverine Ledoux, Cindy Le Bourgot, Pascaline Delaby, Arnaud Bernard, Philippe Besnard

**Affiliations:** 1UMR 1231, INSERM/University of Bourgogne-Franche-Comté/AgroSup Dijon, 21000 Dijon, France; aurelie.dastugue@agrosupdijon.fr (A.D.); arnaud.bernard@u-bourgogne.fr (A.B.); 2Institut du Thorax, CNRS, INSERM, University of Nantes, 44000 Nantes, France; cedric.lemay@univ-nantes.fr; 3Explorations Fonctionnelles, Hôpital Louis Mourier (APHP), 92700 Colombe, France; severine.ledoux@aphp.fr; 4Tereos, 77230 Moussy-le-Vieux, France; cindy.lebourgot@tereos.com; 5Lesieur, 92600 Asnières/Seine, France; pdelaby@lesieur.fr; 6Physiologie de la Nutrition, AgroSup Dijon, 21000 Dijon, France

**Keywords:** obesity, bariatric surgery, gustation, brief-access licking tests, fat and sugar

## Abstract

A preferential consumption of healthier foods, low in fat and sugar, is often reported after bariatric surgery, suggesting a switch of taste-guided food choices. To further explore this hypothesis in well-standardized conditions, analysis of licking behavior in response to oily and sweet solutions has been realized in rats that have undergone a Roux-en-Y bypass (RYGB). Unfortunately, these studies have produced conflicting data mainly due to methodological differences. Paradoxically, whereas the vertical sleeve gastrectomy (VSG) becomes the most commonly performed bariatric surgery worldwide and is easier to perform and standardize in small animals, its putative impacts on the orosensory perception of energy-dense nutrients remains unknown. Using brief-access licking tests in VSG or RYGB mice, we found that (i) VSG induces a significant reduction in the fat mass in diet-induced obese (DIO) mice, (ii) VSG partially corrects the licking responses to lipid and sucrose stimuli which are degraded in sham-operated DIO mice, (iii) VSG improves the willingness to lick oily and sucrose solutions in DIO mice and (iv) RYGB leads to close outcomes. Altogether, these data strongly suggest that VSG, as RYGB, can counteract the deleterious effect of obesity on the orosensory perception of energy-dense nutrients in mice.

## 1. Introduction

The prevalence of morbid obesity (BMI ≥ 40 Kg/m^2^) has increased dramatically in the past decades worldwide, becoming a major public health priority. Due to multiple associated co-morbidities, such as type 2 diabetes, hypertension, cancer, heart and neurodegenerative diseases [1], severe obesity is at the origin of a reduction in the life expectancy ranging from 5 to 20 years [2,3], this trend being all the more dramatic when obesity is precocious [4].

Among the treatments of obesity, bariatric surgery appears to be currently the most efficient method to reduce body weight and correct the associated metabolic disorders in patients with morbid obesity. The different surgery procedures developed have been frequently divided into two categories: the restrictive methods that encompass the adjustable gastric banding, the vertical banded gastroplasty and the vertical sleeve gastrectomy (VSG), and the malabsorptive approaches including the biliopancreatic diversion, the biliointestinal bypass and the Roux-en-Y gastric bypass (RYGB). However, this simplistic distinction proved insufficient to explain the strong variability in the long-term benefits of these surgery procedures, the mechanisms of action being multiple and complex.

Two surgical methods predominate in clinical practice: the VSG and the RYGB. By reason of its long-term effectiveness, RYGB has long been regarded as the bariatric gold standard [5]. This procedure consists of the creation of a small gastric pouch with a new alimentary limb ensured by a Roux-en-Y gastro-jejunostomy, the biliary and pancreatic juice transit being maintained by a jejuno-jejunostomy (biliopancreatic limb), allowing their flow into the remaining small intestine (common channel) [6]. Despite its efficiency, RYGB is gradually being supplanted as the first surgical intervention by VSG that consists of realizing a tube-like structure by removing 75% to 80% of the stomach. This less invasive and challenging procedure combines metabolic benefits and efficient, but sometimes transient, weight loss with a low postoperative complication rate [7]. In 2019, The American Society for Metabolic and Bariatric Surgery (ASMBS) estimated that nearly two-thirds of obesity surgery realized in the USA was VSG.

Paradoxically, how RYGB and VSG work is not fully understood, likely due to the complexity of homeostatic induced changes and the difficulty of linking mechanistic modifications to functional consequences in humans. To overcome these limitations and clarify this question, rat and mouse models of RYGB and VSG have been adapted and generated successfully. The adaptation of these surgeries in mice requires great expertise, high surgical dexterity due to the small size of animals, and adapted specific pre and postoperative protocols of anesthesia/analgesia/nutritional care [8]. From a surgical point of view, the VSG procedure is relatively similar between mice and humans, is well tolerated and the postsurgical survival rate is relatively good in most published studies (reviewed in [9]. However, due to differences in stomach musculature, RYGB is more difficult to transpose from humans to mice and several different models have been described [10]. The mortality rate of the different surgical setups is only reported in 50% of published studies and the average is 29% (reviewed in [9]).

Despite anatomical and physiological differences between rodents and humans (e.g., lack of gallbladder in rats and of the forestomach in humans, species specificities of bile acid metabolism), important functional postoperative similarities were found between these animal models and patients, including not only positive outcomes as fat mass loss, induction of gut satiety hormone secretion, improvement of gut microbiota composition and of insulin sensitivity, but also negative impacts, such as post-RYGB vitamin and iron deficiencies, bone demineralization and alcohol addiction (for review see [9,11]). Although the translational jump from rodent data to human physiology requires caution, these rodent models of bariatric surgery constitute a useful tool in providing mechanistic hypothesis essential to better understand how bariatric surgery corrects homeostatic dysfunctions induced by obesity.

Among the multiple postoperative changes, a preferential consumption for low-caloric foods is often reported by patients who have undergone RYGB or VSG (for reviews, see [12,13,14]). This healthier food selection, which might play a significant role in the success of long-term weight loss, raises the possibility that obesity surgery might also affect the orosensory perception of energy nutrients (i.e., sugar and fat). However, the physiological relevance of this observation is not yet clearly established. Indeed, most of the published clinical data were generated using indirect approaches, such as self-reported questionnaires [14], which can often generate biased responses, especially in patients with obesity [15]. Moreover, the analysis of the taste sensitivity using direct sensory methods, such as alternative forced-choice tests or taste strips, led to discrepant results [14], likely due to the great genetic variability and food habit heterogeneity between patients. To further investigate this question in well-controlled experimental conditions, sweet taste sensitivity has been compared in sham-operated controls and in RYGB rats using brief-access licking tests (e.g., 10 s). The advantage of this behavioral test was to provide information about the taste-driven licking activity (i.e., immediate pleasure or “liking”) and motivation (incentive salience or “wanting”) in response to an oral stimulus independent of post-ingestive cues. Unfortunately, these studies led to controversial outcomes mainly due to methodological differences (Table 1), with some of them reporting a positive correlation between the licking rate and the sucrose concentration, suggesting that RYGB improves sweetness sensitivity [16], while others did not see any change [17,18,19] or even found the opposite result [20,21]. Similarly, the relationship between RYGB and an orosensory perception of dietary lipids also remains unclear, with studies being scarce and conflicting. 

Le Roux et al. found no difference in the Intralipid^TM^ lick scores between RYGB rats and sham-operated controls [22], while Shin et al. reported an improvement in the licking responses to corn oil after RYGB [16]. Finally, although the VSG is also associated with healthier changes in the food selection in rats [23,24,25], the potential impact of this bariatric procedure on the orosensory perception of fat and sweet stimuli is unknown.

The purpose of the present study was to evaluate the respective effects of VSG on the short-term licking responses to fat and sweet stimuli. Brief-access tests were also performed after RYGB in reference to the previously published studies. We have chosen to conduct experiments in the mouse whose multiple transgenic models make possible further studies on the potential mechanisms linking obesity, bariatric surgery and orosensory perception of energy-dense foods. Therefore, the present study was designed according to two complementary goals: explore, for the first time, the effects of VSG on the licking behavior in response to fat and sweet stimuli in obese animals and re-study the role of RYGB on this behavioral parameter, previous data being conflicting. Two distinct cohorts of mice with their own lean and obese controls were used. Bariatric surgery (VSG or RYGB) was performed in diet-induced obese (DIO) mice (VSG-DIO and RYGB-DIO). Since lipid content of diets affects the taste perception independently of body weight changes in rodents [26], mice that underwent VSG or RYGB were maintained on HFD after surgery and were compared to the Sham-DIO controls in order to explore whether the bariatric surgery per se may modify the taste perception.

## 2. Materials and Methods

### 2.1. Animals

Experiments were performed in accordance with the European guidelines for the care and use of laboratory animals. Protocols were approved by the French National Animal Ethics Committee (APAFIS#12728-201712151028893 and APAFIS#19877-2019032109244847). For the estrous cycle affecting the taste reactivity response [27], experiments were only conducted in male mice. Six-week-old C57Bl/6J mice (Charles River Laboratories, Eculie, France) were housed individually in a conventional cage (Eurostandard type II, floor area = 530 cm^2^, Tecniplast, France; bedding material, Scobis Uno, Mucedola SRL, Italia; enrichment: Rodis Uno, Mucedola SRL, Settimo Milanese, Italy) in a controlled environment with a constant temperature (23 ± 1 °C), humidity (60 ± 5%), a dark period from 7 p.m. to 7 a.m., with free access to tap water and chow. Mice were fed either a standard laboratory chow (Sdt-4RF21 autoclaved, 315.0 Kcal/100 g, Mucedola SRL, Settimo Milanese, Italy) or an obesogenic high fat diet (HFD-4RF25 autoclaved, Mucedola SRL, Settimo Milanese, Italy, + 31.8% palm oil, *wt*/*wt*, 505.4 Kcal/100 g). At the end of the experiments, the fat mass was determined by molecular resonance imaging (EchoMRI-Echo Medical Systems, Houston, TX, USA, Figure 1A).

### 2.2. Surgical Procedures and Postoperative Care

The surgical methods used have been fully described and video-captured elsewhere [8]. Prior surgery, all mice received analgesics (buprenorphine 0.1 mg/kg), antibiotics (marbofloxacin 10 mg/kg) and pro-kinetics (metoclopramide 1 mg/kg) via subcutaneous (sc) injections. Anesthesia was induced in a chamber with 5% isoflurane, (air flow of 0.4 L/min and O_2_ flow 0.4 L/min). The VSG procedure consists of removing a large part of the stomach by discarding the forestomach and a part of the corpus (See Figure 1A). Briefly, after performing a median laparotomy, we sutured the pylorus vessels along the stomach’s greater curvature with 8.0 Prolene single sutures to avoid future bleeding. We next performed a resection of the cardiac (forestomach) and pyloric region (corpus) to discard approximately 80% of the stomach. The incision site was sutured with 8.0 Prolene running suture from the gastro-oesophagal origin to the end of the incision. For the RYGB procedure (Figure 1B), after externalization of the first small intestinal loop (duodenum), a double ligature with 5.0 Prolene was realized 4 cm after the end of the duodenum in order to transect the proximal jejunum between the two ligatures. The distal part of the proximal jejunum was moved upon realizing an anastomosis with the stomach which constitutes the alimentary limb. The proximal part of the jejunum was used to perform a side-to-side anastomosis with the median jejunal segment. This Y-shaped anatomical rearrangement leads to a biliopancreatic limb, the remained jejunum constituting the common limb (Figure 1B).

During these two surgery procedures, special care has been taken to avoid any damage to neural and vascular systems, especially at the level of esophagogastric and pyloric areas. The mean operative time for the VSG procedure was 33.5 ± 1.4 min and 60.33 ± 0.88 min for RYGB. Lean and obese controls were sham-operated, taking into account the operative time for each of the surgeries. In our hands, while we have rarely observed deaths following a VSG procedure, the mortality rate in our modified RYGB mouse model varies from 0% to 50%, depending on the training and dexterity of the surgeon. Importantly, all the deaths observed following RYGB occurred in the 5 days after surgery and were mostly due to anastomotic leakage intestinal stenosis. After surgery, all mice received analgesics (buprenorphine 0.1 mg/kg twice daily, from day 0 to day 3 after surgery), antibiotics (marbofloxacin 10 mg/kg from day 0 to day 3 after surgery), and pro-kinetics (metoclopramide 1 mg/kg from day 0 to day 5 after surgery) via sc injections. For gastric bypass inducing vitamin malabsorption and anemia [9], RYGB mice were daily supplemented with poly-vitamins (800 mg/180 mL in drinking water) and iron (0.5 mg/kg/day by sc injection, Figure 1C). Sham-controls received the vehicle alone. For five days after surgery, mice were housed in a 30 °C room and received a gel diet food (Geldiet Highfat, Safe laboratories; lard 10%, liquid sugar 10%, water 57%). Solid food was reintroduced 3 days after surgery.

### 2.3. Brief-Access Licking Tests

Licking tests were performed using an original octagonal-shaped gustometer with eight bottles, each being equipped with a lickometer and a computer-controlled shutter allowing random access during a short time (10 s in the present study) to bottles filled with a specific solution [28]. By forcing the animal to move to have access to the drinking source, this system allows the analysis of licking behavior which mirrors the immediate pleasure gained from the consumption of a rewarding stimulus (i.e., “liking”) and brings information on the motivation to drink (i.e., “wanting”). The concept and procedures are detailed elsewhere [28]. In brief, each of the mice was individually subjected to two training sessions before the taste-testing sessions. The first training is a time of habituation to the apparatus environment with free access to the eight bottles filled with water. Analysis of the number of licks on each bottle provided information about the exploration capacity (curiosity, stress) of the animal, the existence of a bottle location preference, and the rhythmic oromotor activity. During the second training, the mouse learns to drink with random access to the eight bottles as during the taste-testing sessions, with the difference that the only solution available is water. The animal has access to a first bottle for 10 s after the first lick. After this first trial, all shutters remain closed for 10 s before another one is opened in a randomized manner among the seven remaining shutters. The time taken by the mouse to perform the first lick was defined as the latency. Training and taste-testing sessions were computer-controlled to start at the first lick and stop 30 min later. The mouse can initiate as many trials as it wants during this session time. When the mouse has licked the eight bottles (eight trials), it has realized a block. At the end of one block, another block started, so that the number of blocks, and thus of trials, mirrors the motivation for the stimulus. To avoid a memory bias, the shutter opening was designed according to a computer random draw for each block. An animal is excluded from the data if it is unable to realize at least one block in 30 min. To promote licking, mice were overnight water-restricted before each session according to previously published data [18,21,22,29], the experiments being performed in the morning (Figure 1A). To avoid the satiety signals influencing the initial rate of licking [29,30], mice were also food deprived during the dark period before the taste-testing sessions. The restriction led to body mass variations being too limited (−7.7 ± 0.3% of initial body weight on average) to induce physiological disturbances usually associated with dehydration [31]. This body mass change was followed by a nearly full weight regain during the resting time between two successive sessions. No noticeable distress signs were observed in any of the mice. Taste-testing sessions were performed using sucrose (seven concentrations from 0.01 to 0.6 M in water), then rapeseed oil (seven concentrations from 0.03% to 4% in water plus 0.3 xanthan gum to facilitate oil solubilization and minimize textural cues). For each of these sessions, the control solution was the vehicle alone (water alone and water + 0.3% xanthan gum for sucrose and rapeseed oil, respectively).

### 2.4. Statistical Analysis

The statistical tests were performed using R software (v 3.4.4, the R Foundation, Vienna, Austria) with an α level of 0.05. Data not having a normal distribution and variances being not equal, means the difference between groups were analyzed using non-parametric tests (Wilcoxon–Mann–Whitney). The statistical analysis was performed using the obese mice (Sham-DIO) as a reference group, thus providing information on both the efficiency of the obesogenic diet (Sham-L vs. Sham-DIO) and bariatric surgery (Sham-DIO vs. VSG-DIO or RYGB-DIO) on studied parameters. Therefore, lean controls (Sham-L) and obese mice that underwent VSG or RYGB have not been compared since these groups cumulate two types of differences (diet and surgery). A principal component analysis (PCA), normalized and centered, was done with the R-commander package (v2.4.4) using all studied parameters.

## 3. Results

### 3.1. VSG Reduces the Fat Mass despite a Transient Weight Loss

The time-course of the experiment is shown in Figure 1C. Animals were maintained on the same diet throughout the experiment, standard chow for the Sham-L groups and HFD for Sham-DIO, RYGB-DIO, and VSG-DIO animals. Surgery was performed 2 months after the beginning of the protocol. The evolution of body weights after surgery is shown in Figure 1D. By contrast to RYGB, the weight loss after VSG was transient, with mice tending to return to their preoperative weight 10 weeks after surgery (Figure 1E). Despite this difference, VSG as RYGB deeply affected the body composition by reducing the fat mass as compared to their respective sham-DIO controls (Figure 1F).

### 3.2. VSG Improves the Licking Responses to Oily and Sucrose Solutions, as RYGB

In order to study the taste-driven responses to lipid and sweet stimuli, brief-access licking tests (licks/10 s for 30 min) were carried out using an octagonal-shaped gustometer (Figure 2A) according to the protocol shown in Figure 2B. 

The training sessions failed to reveal substantial behavioral differences among the three groups of mice whatever the surgical procedures used, showing that the experimental design did not affect the animals’ adaptability to a new environment (training 1) nor their ability to learn how the device works (training 2, Figure 2C). They also suggest that all the mice displayed similar oromotor and mobility activities.

In Sham-L (VSG or RYGB) mice, the licking activity showed a typical dose–response curve to growing concentrations of oily emulsion (Figure 3A) and sucrose solution (Figure 3B). As previously observed [16], the licking rates decreased dramatically in Sham-DIO mice (VSG or RYGB) as compared to the lean controls (Sham-L), confirming that obese animals are unable to perceive properly fat and sweet stimuli. By contrast, DIO mice that have undergone VSG or RYGB displayed a greater licking activity than their respective obese controls (Sham-DIO). This sensory improvement was substantial, as pointed out by the comparison of both the area under curves (AUC, inserts Figure 3) and the total licks number during the 30 min session. However, this correction was partial, as the licking rates of VSG and RYGB mice remained lower than those found in their respective lean controls (Sham-L).

### 3.3. VSG Improves the Willingness to Lick Oily and Sucrose Solutions in DIO Mice

To explore the motivational component of the taste-driven responsiveness to fat and sweet stimuli, the latency (i.e., the time to initiate the first lick following the access to a solution), and the number of trials (the number of different solutions licked in 30 min) was determined. Latency revealed motivational divergences between groups (Figure 4): the time-response to oily stimuli being greater in obese mice than in lean controls, VSG (and RYGB) animals being in an intermediate position (Figure 4A). While the latency was significantly reduced in the VSG group, only a downward trend was found in the RYGB group (Figure 4A, inserts). Similar changes were also found in response to various sucrose concentrations (Figure 4B). Sham-DIO mice also initiated a fewer number of trials during the 30 min taste-testing session than Sham-L mice, both in response to oily (Figure 4A) or sucrose (Figure 4B) solutions. This relative disinterest was partially reversed in the VSG group, whatever the stimuli tested, but was significant only for sucrose in the RYGB group (Figure 4).

### 3.4. VSG and RYGB Bring Mice Subjected to Obesogenic Diet Closer to Lean Controls

To gain insight into the impacts of the VSG and RYGB procedures in this animal model, a multivariate analysis (principal component analysis—PCA) was performed. The values on the x- and y-axes, describing, respectively, the component score for the dimension 1 and 2, accounted for 68.6 and 11.2% (total = 79.8%) of inertia for the VSG procedure and 64.6 and 14.7% (total, 79.3%) for the RYGB (Figure 5A) demonstrating the existence of strong relationships between studied variables. The confidence ellipse analysis highlighted that obese mice (Sham-DIO) were clearly distinct from the lean controls (Sham-L), whereas the VSG-DIO and RYGB-DIO mice were found in an intermediate position suggesting an improvement of the physiological parameters studied in animals that underwent a bariatric surgery whatever the procedure used. This phenomenon takes place in spite of VSG and RYGB mice being maintained on the obesogenic diet during the course of the study. It is noteworthy that all the mice groups were mainly defined on dimension 1 (Figure 5B).

## 4. Discussion

Most of the patients that have undergone an RYGB, or a VSG reports a change in their orosensory perception of tasty energy-dense foods, a phenomenon usually associated with a preferential consumption of healthier foods, low in sugar and fat (for reviews see, [13,14]). Although these observations suggest a switch in the fat and sweet taste sensitivity, this assumption remains poorly substantiated in humans. To date, the exploration of this issue using rodent models was limited to RYGB rats and has led to discrepant data (Table 1). The present study shows that VSG in mice is a suitable model to better understand how this surgery, commonly performed in humans, can counteract the deleterious effect of obesity on the orosensory perception of energy-dense nutrients observed in rodents [16,32] and re-explores the role of RYGB on this sensory parameter.

Consistent with previous rodent studies [16,28], we found that the licking responses to fat and sweet stimuli were systematically lower in obese mice (Sham-DIO) than in lean controls (Sham-L). Interestingly, this behavioral change was corrected, at least partially, in VSG mice despite they were maintained on HFD after the intervention, showing that this obesity-associated dysfunction is a reversible phenomenon, as previously suggested, using food-restricted DIO mice [32]. For the lipids, our data are congruent with those obtained in RYGB rats by Shin et al. [16] for the same oil concentration range (i.e., from 0.03% to 4%, *w*/*w*). By contrast, they differ from Le Roux et al.’s data [22], showing a positive correlation between the licking rate and Intralipid^TM^ concentrations, both in RYGB and sham-operated rats. The nature of lipids used as a stimulus might partly explain this discrepancy. In contrast to the oil solution used herein and by Shin and co-workers [16], triglycerides (TG) are encapsulated by egg phospholipids in Intralipid^TM^, limiting their hydrolysis by the lingual TG-lipase and, thus, the release of long-chain fatty acids (LCFA). This step is essential for the generation of a lipid signal by the taste bud cells [33]. Therefore, it is likely that the IntralipidT^M^ emulsion is orally perceived, rather through its textural cues via the trigeminal system, than by the LCFA receptors lining the taste bud cells, in contrast to the oily solutions. An improvement of the orosensory perception of sucrose was also observed in VSG mice.

Our data also strongly suggest that VSG modifies the two main components of the orosensory perception of foods, i.e., the hedonic response (“liking”) and the incentive salience (“wanting”) [34]. Indeed, the licking rate, in response to fat and sweet stimuli, which mirrors the pleasure gained from the consumption of a rewarding stimulus, was higher in VSG-DIO mice than in Sham-DIO controls, suggesting again, the orosensory acuity. Moreover, VSG is also associated with a significant change in two indicators of motivation to lick: the time to initiate the first lick during a trial or latency [35] and the number of trials initiated during a session [19]. Indeed, reduced latency and a greater number of trials were found in VSG mice, suggesting a greater motivation to lick oil and sucrose solutions as compared to obese controls. Whether VSG can “reset” the obesity-mediated dysfunction in brain regions processing reward-related behavior, as reported in RYGB rats [36], is not yet known. This point is significant because the reward deficiency observed in DIO animals is thought to be involved in their tendency to overeat high-rewarding fatty foods [16], probably to gain the desired hedonic response [37]. Therefore, the positive outcome of VSG on the orosensory perception of energy-dense nutrients likely contributes to the healthier food choices found in operated mice rats [23,24,25].

In our hands, similar licking changes were also found after RYGB. This last observation substantiates Shin’s data showing that RYGB rats display licking profiles close to lean controls for low concentrations of corn oil and sucrose [16]. However, post-surgery evolution of the body weight is the main difference between the two types of bariatric procedures studied. In contrast to RYGB animals, but as reported in most of the rodent studies [9], VSG mice tend to return to their preoperative weight 10 weeks after surgery, despite a persistent reduction in fat mass. Therefore, the orosensory benefits of VSG in mice might be dependent on the body composition rather than on the long-term weight loss. By releasing pro-inflammatory signaling factors, the white adipose tissue greatly contributes to the obesity-mediated homeostatic disruption, leading to multiple negative functional consequences (e.g., insulin resistance). Using the same cohort of VSG mice, we have recently found that the sham-DIO controls were characterized by a plasmatic neurotoxic signature elicited by the inflammation-mediated overactivation of the tryptophan (Trp) catabolism along the kynurenine (Kyn) pathway. This metabolic change was associated with a reduction in lingual fungiform density and a disturbance of responses to the oily solution during two-bottle preference tests [38]. Interestingly, these metabolic and functional troubles were widely corrected in VSG-DIO mice, suggesting that the Trp/Kyn pathway is one of the factors implicated in the cross-talk between nutritional obesity, bariatric surgery and orosensory perception of lipids. The present data strengthens this previous finding by demonstrating the positive impact of VSG on the licking behavior in response to fat (and sweet) stimuli. Nevertheless, the experimental design used herein has some limits. It does not allow us to determine whether the licks changes found after VSG (and RYGB) results from the surgery per se or is a postoperative consequence (e.g., fat mass loss). In the same way, the fact that VSG and RYGB experiments were performed using two distinct cohorts with their own controls precludes a direct comparison. However, despite significant anatomical and functional species specificities (e.g., lack of forestomach in humans), the main postoperative benefits observed in clinical studies, including the reduction in both fat mass (present study) and insulin resistance [33] were reproduced in VSG mice. Moreover, the plausible involvement of the Trp/Kyn metabolism in the degradation of fatty taste perception identified in the mouse was also observed in a sub-group of patients displaying a significant post-VSG reduction in its inflammatory status [33]. Overall, these data point out that VSG in mice is a suitable model to further explore the molecular mechanisms and functional consequences linking obesity and taste-driven food choices.

In conclusion, this study brings the first demonstration that VSG can counteract, at least partially, the deleterious effects of obesity on the orosensory perception of fat and sweet stimuli and supports the existence of similar action for RYGB. A better understanding of complex mechanisms at the origin of surgery-induced food changes might lead in the future to new strategies targeting the gusto-olfactory system facilitating the long-term compliance with healthy dietary recommendations in patients undergoing bariatric surgery.

## Figures and Tables

**Figure 1 biomedicines-10-00741-f001:**
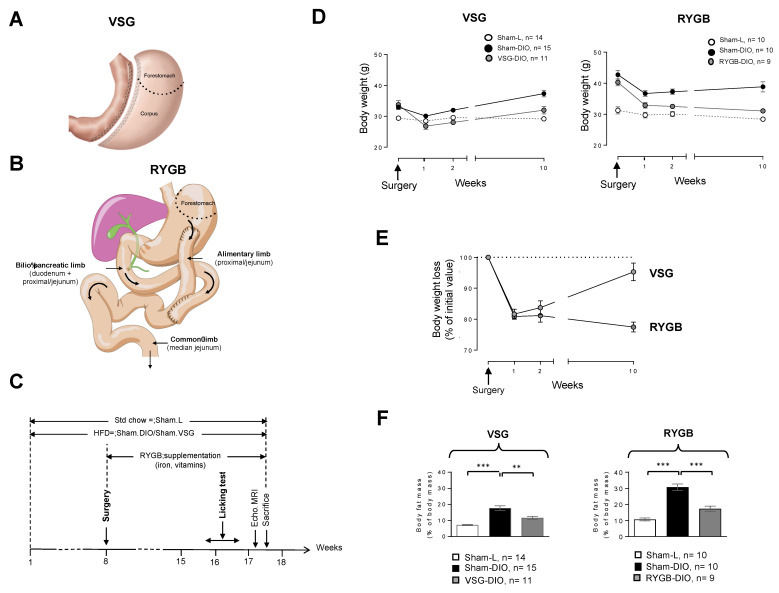
(**A**) Vertical sleeve gastrectomy (VSG) procedure used. (**B**) Roux-en-Y gastric bypass (RYGB) procedure used. (**C**) Time-course of the experiment. (**D**) Evolution of body weight after surgery. (**E**) Evolution of body weight loss after surgery. (**F**) Fat mass at the end of the experiment. Sham-operated lean controls (Sham-L); sham-operated obese controls (Sham-DIO); obese mice that underwent a vertical sleeve gastrectomy (VSG-DIO) or a Roux-en-Y gastric bypass (RYGB-DIO). Mean ± SEM, Wilcoxon and Mann–Whitney tests: ** *p* < 0.01, *** *p* < 0.001. Std—standard laboratory chow; HFD—high fat diet; DIO—diet-induced obesity.

**Figure 2 biomedicines-10-00741-f002:**
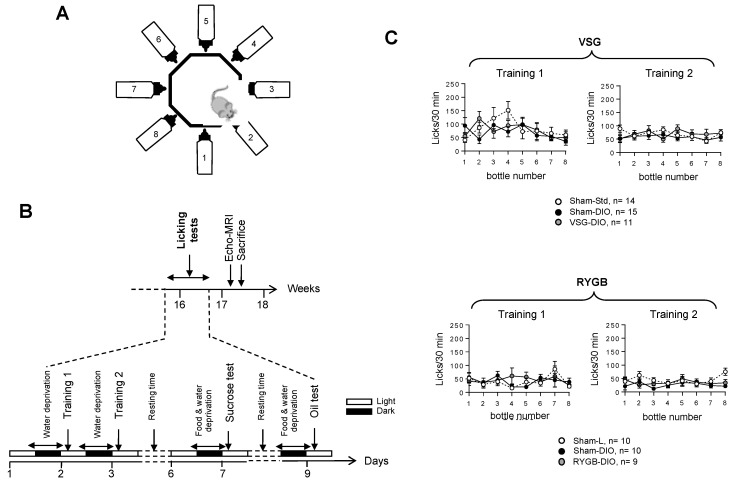
Analysis of the licking behavior in sham-operated lean (Sham-L) and obese (Sham-DIO) controls and in obese mice that underwent a vertical sleeve gastrectomy (VSG-DIO) or a Roux-en-Y gastric bypass (RYGB-DIO) during the training sessions. (**A**) Design of the gustometer. (**B**) Time-course of licking tests session. (**C**) Analysis number of licks per bottle during the training sessions (30 min). Training 1: mice have free access to the eight bottles filled with water. Training 2: mice are subjected to random and intermittent access (10 s) to each of the eight bottles filled with water. Mean ± SEM, Wilcoxon and Mann–Whitney tests. DIO—diet-induced obesity.

**Figure 3 biomedicines-10-00741-f003:**
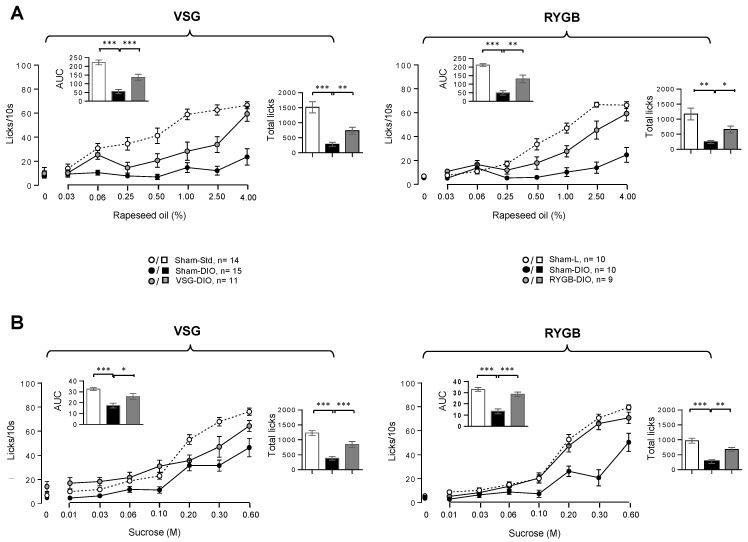
Taste-driven responsiveness to rapeseed oil (**A**) and sucrose (**B**) in sham-operated lean (Sham-L) and obese (Sham-DIO) controls and in obese mice that underwent a vertical sleeve gastrectomy (VSG-DIO) or a Roux-en-Y gastric bypass (RYGB-DIO). Brief-access taste-testing responses (licks/10 s) to various concentrations of the oily emulsion or sucrose solution were presented randomly. Zero on the X-axis represents the licking rate in response to the control solution without oil or sucrose. Analysis of both areas under curves (AUC, Inserts) and total licks during the 30 min of taste-testing sessions were performed using the Sham-DIO group as reference. Mean ± SEM. * *p* < 0.05, ** *p* < 0.01, *** *p* < 0.001. DIO—diet-induced obesity.

**Figure 4 biomedicines-10-00741-f004:**
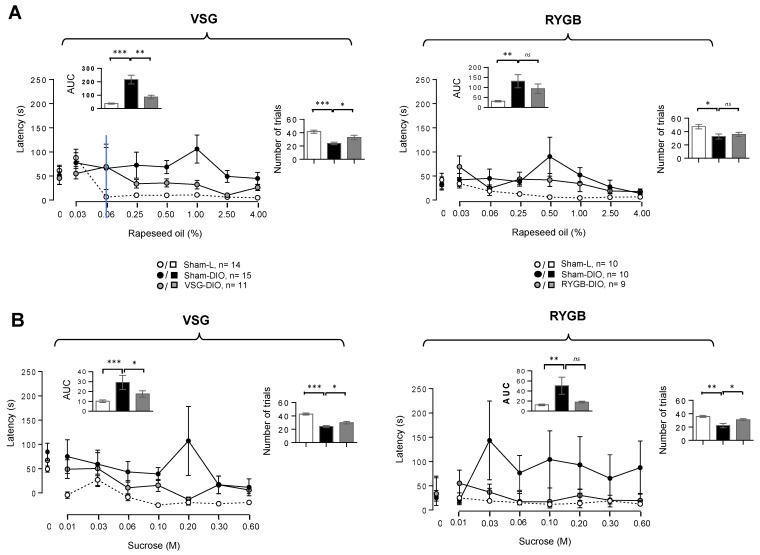
Motivational components of the taste-driven responses to oily emulsion and sucrose solution in sham-operated lean (Sham-L) and obese (Sham-DIO) controls and in obese mice that underwent a vertical sleeve gastrectomy (VSG-DIO) or a Roux-en-Y gastric bypass (RYGB-DIO). Latency is the time between the presentation of a bottle and the initiation of the first lick and the number of trials is the number of solutions licked during the 30 min taste-testing session. (**A**) Oily solutions. (**B**) Sucrose solutions. Zero on the X-axis represents the licking rate in response to the control solution without oil or sucrose. Analysis of both areas under curves (AUC, Inserts) and a number of trials during the 30 min of taste-testing sessions was performed using the Sham-DIO group as reference. Mean ± SEM. *ns*—non-significant, * *p* < 0.05, ** *p* < 0.01, *** *p* < 0.001. DIO—diet-induced obesity.

**Figure 5 biomedicines-10-00741-f005:**
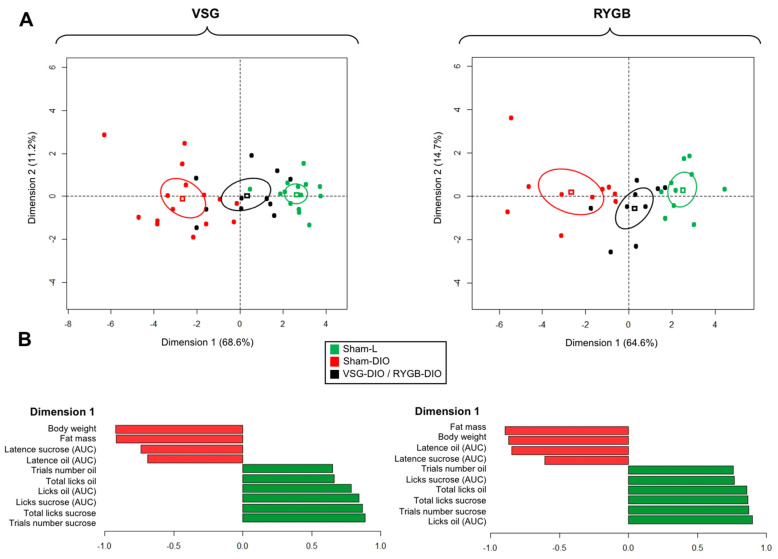
Principal component analysis (PCA) was performed using the studied variables in sham-operated lean (Sham-L) and obese (Sham-DIO) controls and in obese mice that underwent a vertical sleeve gastrectomy (VSG-DIO) or a Roux-en-Y gastric bypass (RYGB-DIO). (**A**) Confidence ellipse analysis with the cluster distribution along the dimension 1 & 2, each dot representing a mouse. (**B**) Main variables characterizing each group of mice.

**Table 1 biomedicines-10-00741-t001:** Impact of RYGB on the taste-driven responses to sweet and fat stimuli in rats.

Strains	Obesity	Surgery	Methods	Findings	References
Male Sprague rats Sham = 11, Surg = 11, Lean = 7	Yes (HFD for 14–16 weeks)	RYGB 20% gastr pouch Alim limb ± 15 cm Bilio-pancr lim ± 40 cm Comm limb ± 25 cm	Licking tests (Davis MS160, trial = 10 s, session = 30 min, 4-6 months after RYGB). Sucrose or corn oil stimuli in random order. Fed animals	RYGB rats display licking profiles close to lean controls for low concentrations of sucrose or corn oil	[16]
Male Sprague rats, Sham = 7, Surg = 7	No (Std diet)	RYGB Gastr pouch remnant Alim limb ± 50 cm Bilio-pancr lim ± 20 cm Comm limb ± 25–30 cm	Licking tests (Davis MS160, trial = 10 s, session = 30 min, 1–1.5 months after RYGB). Sucrose stimuli in random order. Water restriction	No ≠ on the sucrose lick scores between Sham and RYGB groups with or without 23 h fasting	[18]
Female Sprague rats Sham = 7, Surg = 7	No (Std diet)	RYGB Gastr pouch remnant Alim limb ± 50 cm Bilio-pancr lim ± 20 cm Comm limb ± 25–30 cm	Licking tests (Davis MS160 trial = 10 s, session = 30 min, 2–3 months after RYGB). Sucrose stimuli in random order. Water restriction	No attenuation of licking response to a concentration series of sucrose solutions	[19]
Male Long-Evans rats Oletf-RYGB = 6 Pair-fed Oletf-Sham = 4 Leto-RYGB = 6 Pair-fed Leto-Sham = 4	Yes (Genetic CCK-R^–/–^)	RYGB 20% gastr pouch Alim limb ± 10 cm Bilio-pancr lim ± 15 cm Comm limb ± 25–30 cm	Licking tests (Davis MS160 trial = 10 s, session = 20 min, 1 months after RYGB). Sucrose stimuli in random order. Water restriction	RYGB reduces the licking response to high sucrose in OLEF (= obese) rats in contrast to LETO (= lean) rats	[20]
Male Sprague rats Sham = 11, Surg = 9	No (Std diet)	RYGB 10% gastr pouch Alim limb ± 15 cm, Bilio-pancr lim ± 20–25 cm, Comm limb ± 50 cm	Licking tests (Davis MS160 trial = 10 s, session = 30 min, 1 months after RYGB). Sucrose stimuli in random order. Water restriction	RYGB decreases the number of licks for highest sucrose concentrations	[21]
Male Wistar rats Sham = 8, Surg = 8	No (Std diet)	RYGB Gastr pouch remnant Alim limb ± 50 cm Bilio-pancr lim ± 20 cm Comm limb ± 25–30 cm	Licking tests (Davis MS160 trial = 10 s, session = 30 min, 5 months after RYGB). Intralipid stimuli in random order. Water restriction	No ≠ on the Intralipid lick scores between Sham and RYGB groups with or without water restriction	[22]

Sham—sham-operated controls; Std—standard laboratory chow; HFD—high fat diet; RYGB— Roux-en-Y gastric bypass; CCK-1 R^−/−^—cholecystokinin-1 receptor null-mice; Gastr—gastric; Alim—alimentary; Bilio-pancr—bilio-pancreatic; Comm—common.

## Data Availability

Not applicable.

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
