# Peer review of "Taste-Driven Responsiveness to Fat and Sweet Stimuli in Mouse Models of Bariatric Surgery"

_biomedicines, 2022, doi:10.3390/biomedicines10040741_

Round 1

Reviewer 1 Report

The paper "Taste-driven responsiveness to fat and sweet stimuli in mouse models of bariatric surgery " present in vivo study related to the taste perception after bariatric surgery.

The topic is specific and as clearly stated by authors in the introduction and aim this research area need some research due to contradictory results published up to now.

the paper is clear and well presented, cited literature seems sufficient, the overall organization of the manuscript allow to reader to catch all the key points of the research and to evaluate the experimental measuraments

The data are well presented and the results are described in adequate mode. Discussion is appropiate with references to previous published paper and with the authors' comments to their results.

My evaluation on the paper is positive, my main doubt is the fact that also this research add a new series of data in the difficult topic of undestanding the changes of the taste perception of (in this case animals) model animals after gastric surgery. The topic is significant in the area of obesity and of chirurgic treatment of obese. I would suggest to author to enphatise the improvement of their paper compared to the literature also in term of general approach on this issue,

My opinion is of minor comments.

Minor comments on the text can be related to some minor typing mistake please be careful about the naming of groups of the animals in one table the control group is indicated with a different name compared to other tables.

Author Response

Reviewer 1

Minor comments on the text can be related to some minor typing mistake please be careful about the naming of groups of the animals in one table the control group is indicated with a different name compared to other tables.

=> Done (Figures 1-4)

Reviewer 2 Report

Thank you for giving me an opportunity to review this study. The manuscript concerns a relevant and important topic. This present preclinical study entitled “Taste-driven responsiveness to fat and sweet stimuli in mouse models of bariatric surgery” aimed to evaluate the respective effects of vertical sleeve gastrectomy (VSG) on the short-term licking responses to fat and sweet stimuli of C57Bl/6 mice. An in-depth baseline knowledge introduction related to the investigated issue was done clearly by the investigators to guide the readers to the core of research. The manuscript gains its strength as the methods are explained in details for the sake of reproducibility.

  • Please clarify the number (n) of animals for each subgroup of samples collected to be adequate to draw a conclusion with scientific merits.
  • Additionally, I would like to encourage the authors to add a paragraph and report the limitations of the study.
  • I recommend a fine editing and proofreading of the manuscript before publication.

Overall, the manuscript describes a straightforward well-conducted study and is well-written.

Author Response

Reviewer 2 

  • Please clarify the number (n) of animals for each subgroup of samples collected to be adequate to draw a conclusion with scientific merits.

=> Done (Figures 1-4)

  • Additionally, I would like to encourage the authors to add a paragraph and report the limitations of the study.

=> Done (Discussion) : Nevertheless, the experimental design used herein has some limits. It does not allow to determine whether the licks changes found after VSG (and RYGB) results from the surgery per se or is a post-operative consequence (e.g. fat mass loss). In a same way, the fact that VSG and RYGB experiments were performed using two distinct cohorts with their own controls precludes a direct comparison. However, despite significant anatomical and functional species specificities (e.g. lack of forestomach in human) [35] the main post-operative benefits observed in clinical studies including the reduction of both fat mass (present study) and insulin resistance [33] were reproduced in these VSG mice.

  • I recommend a fine editing and proofreading of the manuscript before publication.

 => Done